# The Control of Volume Expansion and Porosity in Carbon Block by Carbon Black (CB) Addition for Increasing Thermal Conductivity

**Min Il Kim [1], Jong Hoon Cho [1,2], Byong Chol Bai [3,\*] and Ji Sun Im [1,4,\*]**

[1]  C1 Gas & Carbon Convergent Research, Korea Research Institute of Chemical Technology (KRICT), Daejeon 34114, Korea; mikim@krict.re.kr (M.I.K.); chojh63@krict.re.kr (J.H.C.)

[2]  Department of Applied Chemical Engineering, Chungnam National University, Daejeon 34134, Korea

[3]  Strategic Planning Division, Korea Institute of Convergence Textile, Iksan 54588, Korea

[4]  Advanced Materials and Chemical Engineering, University of Science and Technology (UST), Daejeon 34113, Korea

\*  Correspondence: baibc820@kictex.re.kr (B.C.B); jsim@krict.re.kr (J.S.I.); Tel.: +82-63-830-3588 (B.C.B.); +82-42-860-7366 (J.S.I.)

**Abstract:** The graphite block as a phase change materials (PCMs) was manufactured by graphitization of a carbon block. Carbon blocks were prepared by filler (cokes or graphite) and binder (pitch). The binder-coated filler was thermally treated for carbonization. The gases generated from the evaporation of low molecular weight components in the binder pitch during the carbonization process were not released to the outside. Consequently, porosity and volume expansion were increased in artificial graphite, and thereby the thermal conductivity decreased. In this study, to prevent the decrease of thermal conductivity in the artificial graphite due to the disadvantages of binder pitch, the carbon block was prepared by the addition of carbon black, which can absorb low molecular weight compounds and release the generated gas. The properties of the prepared carbon blocks were analyzed by SEM, TGA, and thermal conductivity. The addition of carbon black (CB) decreased the porosity and volume expansion of the carbon blocks by 38.3% and 65.9%, respectively, and increased the thermal conductivity by 57.1%. The CB absorbed the low molecular weight compounds of binder pitch and induced the release of generated gases during the carbonization process to decrease porosity, and the thermal conductivity of the carbon block increased.

**Keywords:** Thermal conductivity; volume expansion; carbon black; binder pitch; artificial graphite

## 1. Introduction

Phase change materials (PCMs), which are applied to biotech industries such as temperature-sensitive food transport and biomedical applications, require materials with high thermal conductivity. Graphite has been used as a PCM based on its high thermal conductivity, excellent mechanical properties, good chemical stability, and low thermal expansion [1–3]. Graphite is classified into natural graphite and artificial graphite. The artificial graphite is advantageous to apply to PCMs because it is easy to control its size and properties [4–8].

In general, the raw materials used for artificial graphite are coke filler and binder pitch. Recently, natural graphite (NG) has been used instead of coke filler to prepare carbon blocks owing to its high degree of graphitization and high thermal stability [9–11]. The use of NG circumvents high-temperature graphitization; as a result, it could simplify the preparation process, save energy, and reduce costs [12]. Binder pitch must be used for the preparation of artificial graphite, and is composed of a mixture of various molecular weights compounds. The low molecular weight components in the binder pitch are

volatilized during the carbonization process, leading to the formation of pores. The artificial graphite was expanded by the formed pores, and they decreased the thermal conductivity of the artificial graphite [13–17]; therefore, to increase the thermal conductivity of artificial graphite, it is necessary to control the formed pores by a low molecular weight binder pitch.

Carbon black (CB) consists of solid carbon particles, 10–1000 nm in size, which form aggregates with a grape-cluster shape. The major properties of CB are high absorbent power, high electrical conductivity, antistatic properties, good mechanical reinforcement, and high purity [18–21]. CB was used as an additive to increase the resin content of the binder pitch for an aluminum electrode. CB has been shown to enhance the release of volatile compounds and to act as a nucleation point in polymerization [22–24]. These properties can be applied to improve binder pitch properties.

In this study, carbon blocks were prepared using a binder (pitch) and a filler (natural graphite flakes), and CB was used to control the volume expansion and porosity of the carbonized block. Since the volume expansion and porosity of artificial graphite occurs intensively during the carbonization process, the properties of the carbon block without the graphitization process were investigated. The thermal conductivity of the carbon blocks was evaluated according to the addition of CB, and the variation of the thermal conductivity was considered in terms of porosity, density, and thermal stability. Based on the results, a mechanism underlying the effect of the addition of CB to carbon blocks was derived.

## 2. Materials and Methods

### 2.1. Materials

The raw materials consisted of a filler of natural graphite flakes (NGF, Yanxin Graphite Co., High Purified Flake 99%, Natural Graphite powder, D50 = 44 μm, Pingdu, China), a binder of commercial coal-tar pitch (softening point 112 °C), and an additive of carbon black (CB, Lion Specialty Chemicals Co. Ltd., KB ECP-300J, Tokyo, Japan). The properties of the binder pitch are listed in Table 1.

**Table 1.** The properties of binder pitch.

| Softening Point (°C) | Toluene Insoluble (%) | Quinoline Insoluble (%) | Coking Value (%) | Ash Content (%) |
|---|---|---|---|---|
| 112 | 37 | 8.6 | 65 | 0.25 |

### 2.2. Preparation of Carbon Blocks

To prepare a green block (non-carbonized block), the filler and the binder were mixed at a weight ratio of 7:3 (the conditions of non-cracking on green block), and CB of 0, 1, 5, and 10 phr was added based on the binder pitch. The resulting mixtures were pulverized and mixed for 10 min into a powder by using a pulverizer (700 rpm). The mixed powder was hot-pressed to a solid block ($\varphi 15 \times 3$ mm) with a hot-pressing apparatus. An oil pressure pump was used to provide a unidirectional pressure of 20 MPa to form graphite blocks. The hot-pressing process was carried out at a softening point for 1.5 h. The resulting blocks were then cooled to room temperature. The prepared green blocks were prepared as carbon blocks (after carbonized blocks) through high-temperature thermal treatment at 900 °C. The prepared sample was named C_CB_0 (0 phr carbon black), C_CB_1 (1 phr carbon black), C_CB_5 (5 phr carbon black), and C_CB_10 (10 phr carbon black) according to the content of CB. The prepared carbon blocks were weighed before and after carbonization, and the carbonization yield was measured from Equation (1) based on the measured weight:

$$\text{Carbonization yield} = \left( \frac{\text{Weight of carbonized block}}{\text{Weight of non} - \text{carbonized block}} \right) \times 100 \tag{1}$$

### 2.3. Characterization

A thermo-gravimetric analysis (TGA) was performed to measure the weight loss from the green blocks, NGF, CB, and binder pitch by maintaining a temperature of 900 °C (10 °C/min) in a nitrogen atmosphere. To analyze the density of the prepared green block and carbon block, the helium density and the apparent density were measured. The helium density was measured by using an AccuPyc 1340 (Micromeritics), and the apparent density was measured by the Archimedes method. In addition, the porosity of carbon blocks was calculated using helium density and apparent density. The porosity had been calculated by the following Equation (2) [25]:

$$\text{Porosity} = \left(1 - \frac{\text{Bulk density}}{\text{Real density}}\right) \times 100 \qquad (2)$$

The thermal conductivity of the prepared carbon blocks was measured using a hot disk instrument (TPS2500S, Hot Disk Inc., Göteborg, Sweden) in slab-mode with a sensor type c7577 probe.

## 3. Results and Discussion

### 3.1. Decreased Swelling of Carbon Block According to CB Addition During Carbonization

Images of carbon block shapes before and after carbonization are shown in Figure 1. Non-carbonized blocks showed unmolded powders and cracks. After carbonization, C_CB_0 (the sample of non-CB addition) swelled, and a convex shape carbon block was observed. But the CB added samples did not show a significant difference before and after carbonization. The cross-section of the prepared carbon blocks, shown in Figure 2, was confirmed using SEM. In the SEM images of non-carbonized blocks (a–d), the dark part is the pitch and the bright part is natural graphite flakes (NGF). The non-carbonized blocks had a dense structure in which voids were not found in any of the samples. In the carbonized samples (e–h), the black parts are voids, and the bright parts are the carbonized pitch or NGF. All carbon blocks contained voids, and the void volume was slightly reduced according to CB addition. However, because the SEM image was difficult to quantify, the porosity and volume expansion were used for quantification, as presented in Section 3.4.

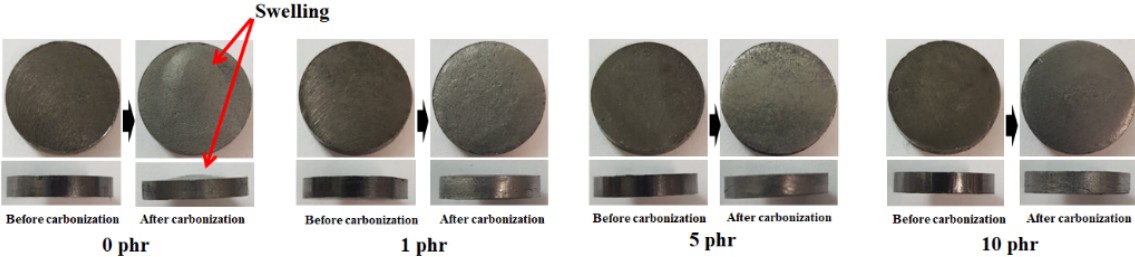

**Figure 1.** Images of before and after carbonized blocks.

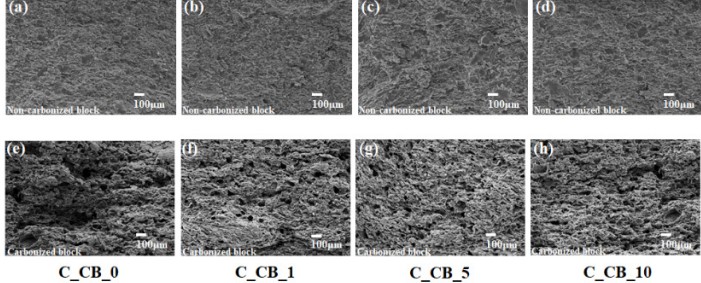

**Figure 2.** SEM images of before and after carbonized blocks; (**a**) non-carbonized C_CB_0, (**b**) non-carbonized C_CB_1, (**c**) non-carbonized C_CB_5, (**d**) non-carbonized C_CB_10, (**e**) carbonized C_CB_0, (**f**) carbonized C_CB_1, (**g**) carbonized C_CB_5, and (**h**) carbonized C_CB_10.

### 3.2. The Thermal Properties of Carbon Block According to CB Addition

The thermal properties of the prepared non-carbonized blocks were evaluated using TGA, as shown in Figure 3. The TGA analysis was carried out from 25 to 900 °C with a heating rate of 10 °C/min in a nitrogen atmosphere. NGF and CB showed a weight loss of 0.8 and 2.7% at 900 °C, respectively. In the case of the pitch, thermal decomposition began near the softening point (112 °C), weight loss did not appear after 550 °C, and the char yield was 44.6%. Thermal decomposition of all samples started at 170 °C, and weight loss did not appear after 550 °C. The char yield of C_CB_0 was 82.6%. On the other hand, all the CB added samples showed increased char yield compared to C_CB_0. The CB had a specific aggregate structure that could absorb low molecular weight compounds in the pitch [26,27]. When the low molecular weight compounds in the pitch were evaporated during the carbonization process, the CB absorbed low molecular weight compounds, and therefore the char yield of the carbon blocks increased. The carbonization yield was measured, and the results are shown in Figure 4. The carbonization yield also showed similar results to the TGA analysis, and it increased by about 2% by the CB addition.

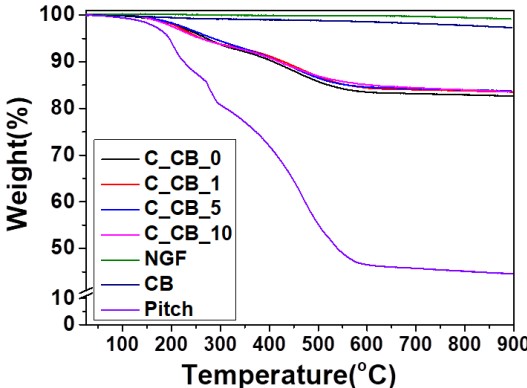

**Figure 3.** TGA spectrum of non-carbonized samples at a heating rate of 10 °C/min in $N_2$.

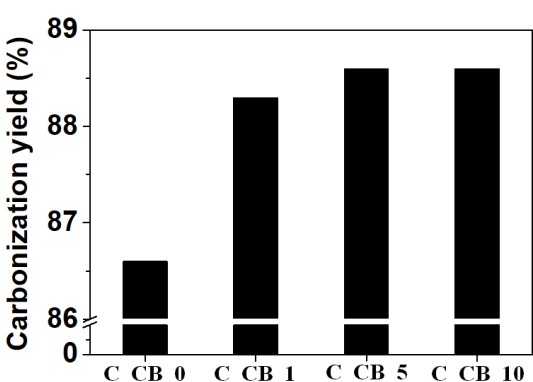

**Figure 4.** The carbonization yield of carbon blocks according to the added CB amount.

### 3.3. The Density of Carbon Blocks According to CB Addition

The density of the prepared carbon blocks was measured using a solid specific gravity meter (bulk density) and a helium measurement method (real density), and the results are shown in Figure 5. The bulk density (Figure 5a) of the non-carbonized C_CB_0 was highest at 2.1 g/cm$^3$, and as the added CB amount increased, the bulk density decreased. It is considered that the bulk density of the non-carbonized block decreased with an increase of the amount of added CB since CB had a low bulk density of 0.125 to 0.145 g/cm$^3$. On the other hand, the carbonized C_CB_0 showed the lowest bulk density among the carbonized blocks (1.51 g/cm$^3$). It is considered that the bulk density of the

carbonized block increased as the amount of added CB increased due to the greater carbonization yield by CB. The bulk density of all samples decreased after carbonization regardless of CB addition.

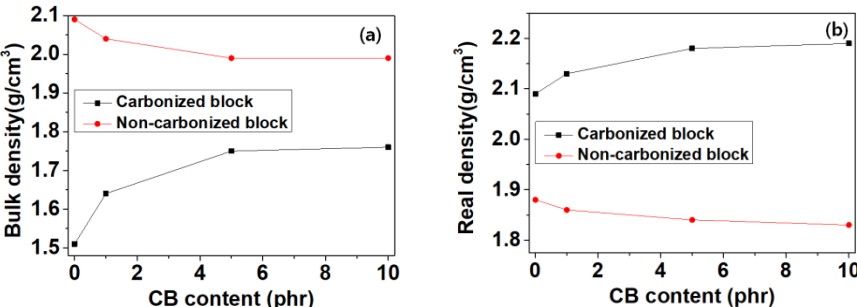

**Figure 5.** The density of carbon blocks according to the added CB amount: (**a**) bulk density and (**b**) real density.

The real density of the prepared carbon blocks is shown in Figure 5b. According to the increase of the amount of added CB, the real density of the non-carbonized block decreased and the real density of the carbonized block increased. The real density of all samples increased after carbonization regardless of CB addition.

### 3.4. Volume Expansion and Porosity Control of Carbon Block by CB Addition

The volume expansion of blocks by the carbonization process was calculated from the bulk density, and the results are shown in Figure 6. As the amount of added CB increased, volume expansion of the carbon blocks decreased. In general, the carbon block was expanded by gases generated from the evaporation of low molecular weight compounds contained in the pitch during the carbonization process [28–30]. The carbon block with coal-tar pitch (binder pitch) also showed volume expansion, and the volume expansion of C_CB_0 was the highest (38.4%) among all samples. The addition of CB decreased the volume expansion; C_CB_10 showed the lowest volume expansion at 13.1%. The decrease of volume expansion of carbon block, according to CB addition, could be confirmed through two aspects. First, as shown in the results of the thermal properties of carbon blocks, the carbonization yield increased by CB absorbing the low molecular weight compounds in the binder pitch during the carbonization process [26,27]. These results were due to the oil absorption ability of the CB, and the volume expansion of carbon block could be partially inhibited by preventing the evaporation of low molecular weight compounds. Second, the CB had a specific aggregate structure and this structure could prevent the volume expansion of carbon block by the release of generated gases to the outside [31]. The porosity was calculated using the bulk density and real density of the carbonized carbon block, and the results are shown in Figure 7. The porosity of the carbonized block also decreased according to increasing of CB amount. In the carbonization process, the binder pitch is changed to a dense structure. However, in cases where the generated gas from binder pitch was not released to the outside and remained inside, the generated gas formed pores and the porosity of block increased [32]. CB absorbed low molecular weight compounds of binder pitch during carbonization to reduce generated gas. Also, CB released gas to the outside and suppressed the generation of pores. Therefore, as the amount of CB added increased, the porosity of the block decreased [31].

The mechanism was defined based on the results of the CB addition to the carbon blocks, and is suggested in Figure 8. The CB could absorb the low molecular weight compounds in the binder pitch during the carbonization process [26,27]. The properties of CB reduced the amount of gas generated by suppressing the evaporation of low molecular weight compounds (Figure 8b). In general, CB had a unique bundle structure similar to grape clusters, and CB's unique bundle structure could diffuse gas, so CB was mainly applied to gas diffusion electrodes [33–35]. A pass-way to release gas was created by the gas diffusion effect of CB in carbon block. The created pass-way released evaporated

low molecular weight compounds to the outside, preventing volume expansion and reducing porosity due to gas inside the block (Figure 8c), so, as the amount of CB added increased, the volume expansion and porosity of the carbon block decreased, because the CB bundles were adjacent to each other and could release a greater amount of gas to the outside.

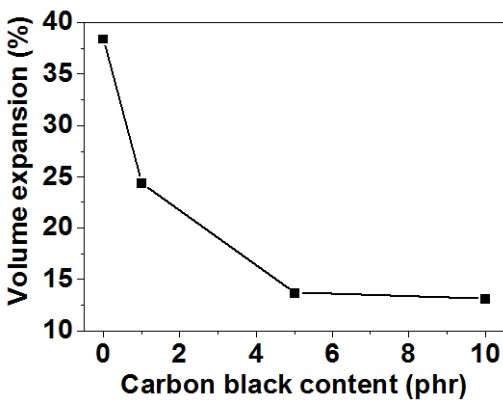

**Figure 6.** The volume expansion of carbon blocks according to the added CB amount.

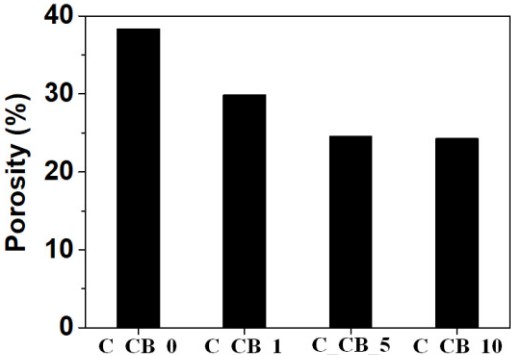

**Figure 7.** The porosity of carbon blocks according to the added CB amount.

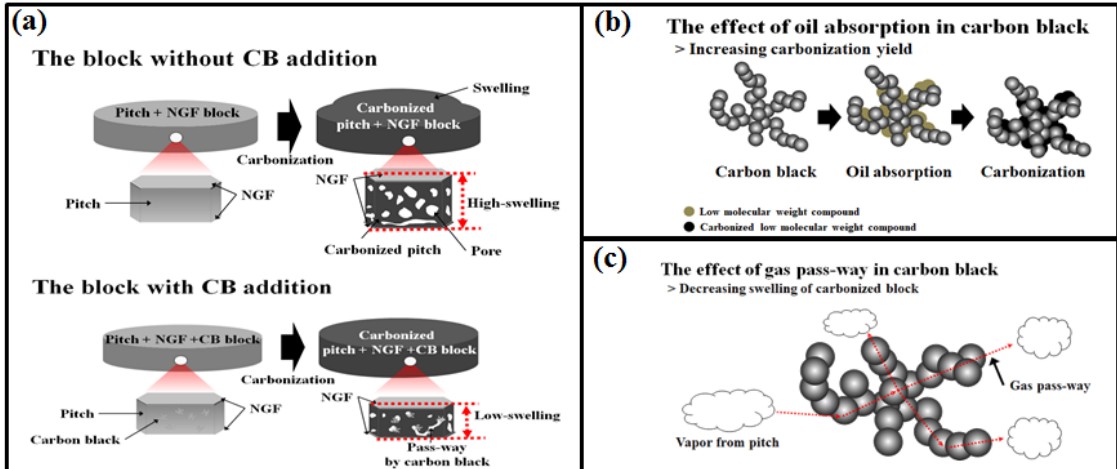

**Figure 8.** The suggested mechanism for the effect of added CB in carbon block during the carbonization process; (**a**) the change of block shape by CB addition, (**b**) the effect of oil absorption in CB, and (**c**) the effect of gas pass-way in CB.

### 3.5. Improvement of Thermal Conductivity in Carbon Blocks According to CB Addition

The thermal conductivity of the prepared carbon blocks was analyzed by the hot-disk method and the results are shown in Figure 9. To prepare a high thermal conductivity carbon block, large particle size NGF should be used, but in this study, NGF with small particle size (44 μm) was used as a filler to confirm the effect of carbon black addition. The CB addition increased the thermal conductivity of the carbon block, but the thermal conductivity did not show a significant change even when the amount of CB added increased. In general, as the volume expansion and porosity of the carbon block decreased, the thermal conductivity increased [36–39]. Therefore, the thermal conductivity of C_CB_10 (the lowest volume expansion and porosity) should be the highest, but the thermal conductivity of C_CB_10 was not significantly different from C_CB_1 and C_CB_5. CB had a low thermal conductivity of less than 2 W/mK [40]. As the CB amount with low thermal conductivity increased, the thermal conductivity of the carbon block decreased. As a result, as the CB amount increased, the thermal conductivity of the carbon block was increased by reducing the volume expansion and porosity of the carbon block. But due to the low thermal conductivity of CB, the thermal conductivity of the carbon block did not change significantly even when the content of CB increased.

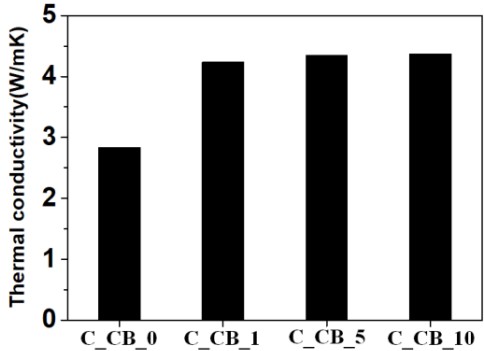

**Figure 9.** The thermal conductivity of carbon blocks according to the added CB amount.

## 4. Conclusions

Carbon blocks were prepared by binder pitch and natural graphite flakes, and CB was added to increase the thermal conductivity for the application of biomedical applications. CB was added to the mixing process with binder pitch and NGF to prepare a mixture for carbon blocks. The carbon blocks were prepared from a mixture through hot-press and carbonization. The properties of the carbon blocks were analyzed by thermal conductivity, SEM, TGA, bulk density, and real density according to the amount of added CB. The CB had an oil absorption property and a unique grape-cluster structure. The CB addition increased the carbonization yield of carbon block by absorption of the low molecular weight compounds from the binder pitch during the carbonization process. Also, the specific aggregate structure of CB could prevent volume expansion of the carbon block by the release of generated gases to the outside. The CB decreased volume expansion of the carbon blocks from 38.4 to 13.1%, and the porosity also decreased from 38.4 to 23.3%. As a result, the thermal conductivity of the carbon block increased from 2.8 W/mK to 4.4 W/mK due to the decrease in volume expansion and porosity by CB addition. However, due to the low thermal conductivity of CB, the thermal conductivity did not show a significant change even when the CB content was increased. These research results are expected to be applicable to the manufacturing of high-thermal conductivity graphite blocks for PCM.

**Author Contributions:** Conceptualization, M.I.K., B.C.B., and J.S.I.; data curation, M.I.K. and B.C.B.; formal analysis, J.H.C.; funding acquisition, J.S.I.; investigation, M.I.K. and J.H.C.; project administration, B.C.B.; resources, J.S.I.; supervision, J.S.I.; validation, J.H.C. and B.C.B.; visualization, M.I.K. and B.C.B.; writing–original draft, M.I.K.; writing–review and editing, J.S.I. and B.C.B.; All authors have read and agreed to the published version of the manuscript.

**Funding:** This research was funded by Korea Evaluation Institute of Industrial Technology, grant number 10083621.

**Acknowledgments:** This work was supported by the Korea Evaluation Institute of Industrial Technology (KEIT) through the Carbon Cluster Construction project [10083621, Development of Preparation Technology in Petroleum-Based Artificial Graphite Anode funded by the Ministry of Trade, Industry, and Energy (MOTIE, Korea)].

**Conflicts of Interest:** The authors declare no conflict of interest. The funders had no role in the design of the study; in the collection, analyses, or interpretation of data; in the writing of the manuscript, or in the decision to publish the results.

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
