# Peer review of "The Control of Volume Expansion and Porosity in Carbon Block by Carbon Black (CB) Addition for Increasing Thermal Conductivity"

_applsci, doi:10.3390/app10176068_

Round 1

Reviewer 1 Report

This manuscript presents the research of adding carbon black into carbon blocks to suppress the volume expansion during carbonization. The knowledge could be technically relevant for a specific process. However, there seem no clearly new or significant results in the research to warrant a publication. Even after adding carbon black, the maximum thermal conductivity of the graphite-based carbon blocks only increases from 2.8 W/m K to 4.4 W/m K, much lower than other reported values [see e.g., ~ 200 W/m K in Carbon 50, 3947-3958 (2012)]. In addition, the “mechanism” discussion about the prevention of volume expansion by carbon black additives, i.e., the aggregate structure, seems merely based on speculation, without any solid evidence or relevant characterization.

Author Response

Q. This manuscript presents the research of adding carbon black into carbon blocks to suppress the volume expansion during carbonization. The knowledge could be technically relevant for a specific process. However, there seem no clearly new or significant results in the research to warrant a publication. Even after adding carbon black, the maximum thermal conductivity of the graphite-based carbon blocks only increases from 2.8 W/m K to 4.4 W/m K, much lower than other reported values [see e.g., ~ 200 W/m K in Carbon 50, 3947-3958 (2012)]. In addition, the “mechanism” discussion about the prevention of volume expansion by carbon black additives, i.e., the aggregate structure, seems merely based on speculation, without any solid evidence or relevant characterization.

A. We agree that the prepared carbon block has a lower thermal conductivity compared to the other research results. Also, various new methods had been introduced to prepare carbon blocks with high thermal conductivity. However, there is also a need to improve traditional artificial graphite manufacturing methods. The purpose of this study is to control the volume expansion and porosity of the carbon block in the artificial graphite manufacturing method. Therefore, rather than the absolute value of the thermal conductivity, it was focused on the improvement rate of the thermal conductivity in the carbon block by the addition of carbon black. The improvement rate of thermal conductivity is 57.1% by CB addition; it is considered a meaningful result. In addition, the mechanism was investigated by adding related references and added as follows.

“The CB could absorb the low molecular weight compounds in the binder pitch during the carbonization process [26, 27]. The properties of CB reduced the amount of gas generated by suppressing the evaporation of low molecular weight compounds (Fig. 8 (b)). In general, CB had a unique bundle structure similar to grape clusters, and CB's unique bundle structure could diffuse gas, so CB was mainly applied to gas diffusion electrodes [33-35]. A pass-way to release gas was created by gas diffusion effect of CB in carbon block. The created pass-way was released evaporated low molecular weight compounds to the outside, preventing volume expansion and reducing porosity due to gas inside the block (Fig. 8 (c)). So, as the amount of CB added increased, the volume expansion and porosity of the carbon block decreased, because the CB bundles were adjacent to each other and could release a greater amount of gas to the outside.”

Reviewer 2 Report

The manuscript presents new results and has acceptable quality.

Recommendation: Accept after minor revisions.

Comments:

p.1, Abstact, line 23 - reformulate. Surely, The properties of the material can be analyzed by SEM and TGA, while density and thermal conductivity can effect on the other properties, but the properties cannot be analized by density.

p.7, Conclusion, lines 190-191 - reformulate. A similar comment.

p.1, Abstract, lines 28-29 - reformulate.

p.3, lines 106-107 - reformulate (the samples with and without the addition of carbon black?).

Author Response

Q. p.1, Abstact, line 23 - reformulate. Surely, The properties of the material can be analyzed by SEM and TGA, while density and thermal conductivity can effect on the other properties, but the properties cannot be analized by density.
A. The density was removed from the analysis method.
“The properties of the prepared carbon blocks were analyzed by SEM, TGA, and thermal conductivity.”

Q. p.7, Conclusion, lines 190-191 - reformulate. A similar comment.
A. The sentence has been modified according to the reviewer's comment.
“CB was added to the mixing process with binder pitch and NGF to prepare a mixture for carbon blocks. The mixture was prepared with carbon blocks through hot-press and carbonization.”

Q. p.1, Abstract, lines 28-29 - reformulate.
A. The sentence has been deleted.

Q. p.3, lines 106-107 - reformulate (the samples with and without the addition of carbon black?).
A. The sentence has been modified according to the reviewer's comment.
“Images of carbon block shapes before and after carbonization are shown in Fig. 1. Non-carbonized blocks showed unmolded powders and cracks.”

Reviewer 3 Report

This paper investigates the effect of carbon black (CB) addition in carbon block fabrication on the thermal conductivity, from the viewpoints of volume expansion during carbonization and porosity. The content of experiment is clear and understandable, but the abstract and the order of data showing is confusing. So I think this paper needs major revision.

I commented for the manuscript as follows.

Title

“for biomedical materials”

It is not suitable. No biomedical data exhibited in this paper.

Abstract

Very confusing. I could not catch the content and importance of this paper at a glance.

What is investigated? (Why investigated?) By what methods? What is the essential results? This paper contributes to what?

Author should improve.

L24

CB is firstly shown.

LL36-37

“Graphite has been used as a PCMs based on their high thermal conductivity, excellent mechanical properties, good chemical stability, and low thermal expansion.”

Please add some references for readers to understand why graphite is necessary to PCMs.

L74

“7:3”

Why this condition is selected?

L74

“carbon black of 0, 1, 5, and 10 phr”

There is no information of the sample number in manuscript.

I think a list of the sample condition is necessary.

L90 Result

In experimental section, thermal conductivity measurement is written at the end, while the result of thermal conduction is located at the first. This is very strange and confusing.

I recommend the section of thermal conductivity should be located at the end of results.

L96

“C_CB_5 showed the highest thermal conductivity”

Why the sample of 5% is highest?

The thermal conductivity is nearly the same for 1%, 5%, and 10%. Authors should improve the explanation and please mention the reason why the conductivity is nearly the same?

Fig. 2

The swelling of the sample seems important, so add cross-sectional view to show the swelling dilectly.

L131

“carbonization yield was measured”

By what?

L171

“The porosity was calculated”

Please show the equation to calculate the porosity to clarify the physical meaning of the porosity.

L172-173

“The decreased porosity is also considered to be an effect of the CB.”

Is this "an effect?"

Two reasons are shown above. Which one? Or “effects?”

LL178-186

Please improve this section. Too little information in this section.

Ex) the mechanism of gas pass-way should be mentioned.

Fig. 9

In right lower figure, why CB works as gas pass-way? is there any evidence?

Author Response

Q. [Title] “for biomedical materials” It is not suitable. No biomedical data exhibited in this paper.
A. The title has been modified according to the reviewer's comment.
“The control of volume expansion and porosity in carbon block by carbon black (CB) addition for increasing thermal conductivity”

Q. [Abstract] Very confusing. I could not catch the content and importance of this paper at a glance. What is investigated? (Why investigated?) By what methods? What is the essential results? This paper contributes to what? Author should improve.
A. The abstract has been modified as follows taking according to the reviewer's comment.
“The graphite block as a phase change materials (PCMs) was manufactured by graphitization of a carbon block. Carbon blocks were prepared by filler (cokes or graphite) and binder (pitch). The binder-coated filler was thermally treated for carbonization. The gases generated from the evaporation of low molecular weight components in binder pitch during the carbonization process were not released to the outside. Consequently, porosity and volume expansion was increased in artificial graphite, and thereby the thermal conductivity decreased. In this study, to prevent the decrease of thermal conductivity in artificial graphite due to the disadvantages of binder pitch, the carbon block was prepared by the addition of carbon black, which can absorb low molecular weight compounds and release the generated gas. The properties of the prepared carbon blocks were analyzed by SEM, TGA, and thermal conductivity. The addition of CB decreased the porosity and volume expansion of the carbon blocks by 38.3% and 65.9%, respectively, and increased the thermal conductivity by 57.1%. The CB absorbed the low molecular weight compounds of binder pitch and induced the release of generated gases during the carbonization process to decrease porosity, and the thermal conductivity of the carbon block increased.”

Q. [L24] CB is firstly shown.
A. The wrong abbreviation was modified.
“the carbon block was prepared by the addition of carbon black (CB)”

Q. [LL36-37] “Graphite has been used as a PCMs based on their high thermal conductivity, excellent mechanical properties, good chemical stability, and low thermal expansion.” Please add some references for readers to understand why graphite is necessary to PCMs.
A. References have been added.
“Graphite has been used as a PCMs based on their high thermal conductivity, excellent mechanical properties, good chemical stability, and low thermal expansion [1-3].”
[1] Opolot, M.; Zhao, C.; Liu, M.; Mancin, S.; Bruno, F; Hooman, K. Influence of cascaded graphite foams on thermal performance of high temperature phase change material storage systems. Appl. Therm. Eng. 2020, 180, 115618.
[2] Luo, D.; Xiang, L.; Sun, X.; Xie, L.; Zhou, D.; Qin, S.; Phase-change smart lines based on paraffin-expanded graphite/polypropylene hollow fiber membrane composite phase change materials for heat storage. Energy 2020, 197, 117252.
[3] Luo, X.; Guo, Q.; Li, X.; Tao, Z.; Lei, S.; Liu, J.; Kang, L.; Zheng, D.; Liu, Z. Experimental investigation on a novel phase change material composites coupled with graphite film used for thermal management of lithium-ion batteries. Renew. Energy 2020, 145, 2046-2055.

Q. [L74] “7:3” Why this condition is selected?
A. As a result of previous research on carbon block molding, cracking occurred when the pitch ratio was low. The ratio of 7:3 was selected because it is a condition for the prevention of crack in a block when carbon black adds.
“To prepare a green block (non-carbonized block), the filler and the binder were mixed at a weight ratio of 7:3 (the conditions of non-cracking on green block), and CB of 0, 1, 5, and 10 phr was added based on the binder pitch.”

Q. [L74] “carbon black of 0, 1, 5, and 10 phr” There is no information of the sample number in manuscript. I think a list of the sample condition is necessary.
A. The a list of the sample condition was added.
“The prepared sample was named C_CB_0 (0 phr carbon black), C_CB_1 (1 phr carbon black), C_CB_5 (5 phr carbon black), and C_CB_10 (10 phr carbon black) according to the content of CB.”

Q. [L90 Result] In experimental section, thermal conductivity measurement is written at the end, while the result of thermal conduction is located at the first. This is very strange and confusing. I recommend the section of thermal conductivity should be located at the end of results.
A. The thermal conductivity section has been modified to locate at the end of the results.

Q. [L96] “C_CB_5 showed the highest thermal conductivity” Why the sample of 5% is highest?
The thermal conductivity is nearly the same for 1%, 5%, and 10%. Authors should improve the explanation and please mention the reason why the conductivity is nearly the same?
A. A discussion has been added for the reason that the similar thermal conductivity of the samples.
“The CB addition increased the thermal conductivity of the carbon block, but the thermal conductivity did not show a significant change even when the amount of CB added increased. In general, as the volume expansion and porosity of the carbon block decrease, the thermal conductivity increases [36-39]. So the thermal conductivity of C_CB_10 (the lowest volume expansion and porosity) should be the highest, but the thermal conductivity of C_CB_10 was not significantly different from C_CB_1 and C_CB_5. As a result, as the CB amount increased, the thermal conductivity of the carbon block was increased by reducing the volume expansion and porosity of the carbon block. But due to the low thermal conductivity of CB, the thermal conductivity of the carbon block did not change significantly even when the content of CB increased. Because CB had a low thermal conductivity of less than 2 W/mK [40].”

[40] Hauser, R.A.; King, J.A.; Pagel, R.M.; Keith, J. M. Effects of carbon fillers on the thermal conductivity of highly filled liquid-crystal polymer based resins. J. Appl. Polym. Sci. 2008, 109, 2145-2155.

Q. [Fig. 2] The swelling of the sample seems important, so add cross-sectional view to show the swelling dilectly.
A. The cross-sectional view of samples was added.

Q. [L131] “carbonization yield was measured” By what?
A. The equation (1) for carbonization yield has been added.

Q. [L171] “The porosity was calculated” Please show the equation to calculate the porosity to clarify the physical meaning of the porosity.
A. The equation (2) for porosity of carbon block has been added.

Q. [L172-173] “The decreased porosity is also considered to be an effect of the CB.”
Is this "an effect?" Two reasons are shown above. Which one? Or “effects?”
A. The discussion of “effect of the CB” has been added.
“The porosity of the carbonized block also decreased according to increasing of CB amount. In the carbonization process, the binder pitch is changed a dense structure. However, in case of the generated gas from binder pitch was not released to the outside and remained inside, generated gas formed pore, and the porosity of block increased [32]. CB absorbed low molecular weight compounds of binder pitch during carbonization to reduce generated gas. Also, CB released gas to the outside and suppressed the generation of pores. Therefore, as the amount of CB added increases, the porosity of the block decreased [31].”

Q. [LL178-186] Please improve this section. Too little information in this section.
Ex) the mechanism of gas pass-way should be mentioned.
[ Fig. 9] In right lower figure, why CB works as gas pass-way? is there any evidence?
A. The discussion of mechanism section has been added.
“The mechanism was defined based on the results of the CB addition to the carbon blocks, and is suggested in Fig. 8. The CB could absorb the low molecular weight compounds in the binder pitch during carbonization process [26, 27]. The properties of CB reduced the amount of gas generated by suppressing the evaporation of low molecular weight compounds (Fig. 8 (b)). In general, CB had a unique bundle structure similar to grape clusters, and CB's unique bundle structure could diffuse gas, so CB was mainly applied to gas diffusion electrodes [33-35]. A pass-way to release gas was created by gas diffusion effect of CB in carbon block. The created pass-way was released evaporated low molecular weight compounds to the outside, preventing volume expansion and reducing porosity due to gas inside the block (Fig. 8 (c)). So, as the amount of CB added increased, the volume expansion and porosity of the carbon block decreased, because the CB bundles were adjacent to each other and could release a greater amount of gas to the outside.”

Round 2

Reviewer 1 Report

The authors provide reasonable explanation for the low thermal conductivity of carbon block and more reference and background to support the proposed mechanism. I think the version is acceptable for publication.

Reviewer 3 Report

The manuscript is well improved. 

I think this paper is suitable to accept in present form.